# Development and comparison of loop-mediated isothermal amplification with quantitative PCR for the specific detection of *Saprolegnia* spp.

**Satyaki Ghosh**[1¤], **David L. Straus**[2], **Christopher Good**[3], **Vipaporn Phuntumart**[1]*

**1** Department of Biological Sciences, Bowling Green State University, Bowling Green, Ohio, United States of America, **2** United States Department of Agriculture, Agricultural Research Service, Harry K. Dupree-Stuttgart National Aquaculture Research Center, Stuttgart, Arkansas, United States of America, **3** The Conservation Fund's Freshwater Institute, Shepherdstown, West Virginia, United States of America

¤ Current address: AgriPlex Genomics, Cleveland, Ohio, United States of America
* vphuntu@bgsu.edu

**Data Availability Statement:** All DNA sequences are available from the NCBI database: BankIt2510419 N9 OK872234 BankIt2510427 S. salmonis OK872235 BankIt2510436 F1 OK872236 BankIt2510443 F2 OK872237 BankIt2510448

## Abstract

Saprolegniasis is an important disease in freshwater aquaculture, and is associated with oomycete pathogens in the genus *Saprolegnia*. Early detection of significant levels of *Saprolegnia* spp. pathogens would allow informed decisions for treatment which could significantly reduce losses. This study is the first to report the development of loop-mediated isothermal amplification (LAMP) for the detection of *Saprolegnia* spp. and compares it with quantitative PCR (qPCR). The developed protocols targeted the internal transcribed spacer (ITS) region of ribosomal DNA and the cytochrome C oxidase subunit 1 (CoxI) gene and was shown to be specific only to *Saprolegnia* genus. This LAMP method can detect as low as 10 fg of *S. salmonis* DNA while the qPCR method has a detection limit of 2 pg of *S. salmonis* DNA, indicating the superior sensitivity of LAMP compared to qPCR. When applied to detect the pathogen in water samples, both methods could detect the pathogen when only one zoospore of *Saprolegnia* was present. We propose LAMP as a quick (about 20–60 minutes) and sensitive molecular diagnostic tool for the detection of *Saprolegnia* spp. suitable for on-site applications.

## Introduction

Oomycetes represent a diverse group of eukaryotic pathogens that can infect a wide range of hosts [1–3] Pathogens in the *Saprolegnia* genus can infect and kill crustaceans and fish in freshwater, especially in aquaculture where fish are raised at high densities [4]. The most widely studied members of this genus include *S. diclina*, *S. ferax* and *S. parasitica* [5].

Saprolegniasis has been reported in channel catfish (*Ictalurus punctatus*), rainbow trout (*Oncorhynchus mykiss*), redear sunfish (*Lepomis microlophus*), Indian major carps (including *Labeo rohita*, *Catla catla*, *L. calbasu*) and Atlantic salmon (*Salmo salar*) [6–8]. The majority saprolegniasis occurrence has been reported in temperate environments and commonly found associated with salmonid farming. *Saprolegnia* can infect all stages of the fish life cycle

JMM_Shad_3 OK872238 BankIt2510453 1B OK872239 BankIt2510454 6A OK872240 BankIt2510456 11A OK872241 BankIt2510457 RAS_1 OK872242 BankIt2510459 RAS_2 OK872243 BankIt2510461 RAS_3 OK872244 BankIt2510463 RAS_4 OK872245 BankIt2510466 RAS_5 OK872246 BankIt2511910 N3 OK872247 BankIt2510450 JMM_Shad_72 OK872248.

**Funding:** SG-Ohio Sea Grant (60063782), https://ohioseagrant.osu.edu/ VP-Ohio Sea Grant (60063782), https://ohioseagrant.osu.edu/ The funders had no role in study design, data collection and analysis, decision to publish, or preparation of the manuscript.

**Competing interests:** The authors have declared that no competing interests exist.

including the egg stage [9–11]. In the U.S., economic losses due to saprolegniasis in commercial aquaculture is estimated to be approximately 40 million dollars [12, 13]. Almost 50% of the commercial losses to farmed channel catfish is attributed to saprolegniasis [14]. It is estimated that 10% of all hatched salmon succumb to saprolegniasis, which represents a major financial loss given that this industry accounts for approximately 30% of the global fish production for consumption [4, 14–16]. The incidence of saprolegniasis extends to Asian tropical aquaculture systems where over 80% of fish is produced through aquaculture [17]. Therefore, saprolegniasis is among the most impactful diseases in commercial aquaculture.

Conventional culture-based strategies coupled with conventional PCR-based approaches for pathogen identification usually take several days [18]. Currently, quantitative-PCR (qPCR) is one of the methods of choice for molecular detection and quantification of target organisms [19]. Detection of pathogens based on qPCR has been reported in several studies including *Trypanosoma cruzi*, *Fusarium* spp., Japanese encephalitis virus and bacteria such as *Listeria monocytogenes*, *Francisella tularensis*, and *Mycobacterium avium* [20–23].

Loop-Mediated Isothermal Amplification (LAMP) has emerged as a novel tool for the sensitive and rapid detection of nucleic acids and is suitable for field applications. Since the first publication of LAMP [24], there has been a progressive increase in the number of publications based on this technique to diagnose and detect a variety of organisms including bacteria, viruses and eukaryotic pathogens indicating the success of LAMP as a reliable tool for detection [25–27]. The present focuses on development of a LAMP and a qPCR method for specific detection of *Saprolegnia* species, and compared these two tests based on sensitivity and specificity. These two molecular approaches targeted the internal transcribed spacer (ITS) region of ribosomal DNA and the mitochondrial cytochrome c oxidase subunit I (C0xI) gene. We first report the standard protocols for specific detection of the pathogens, followed by testing our methods under various scenarios for detection of *Saprolegnia* species including water samples, infected fish tissue, mycelium and zoospores. Our results show that both of the methods are reliable, specific and sensitive to *Saprolegnia*, and have the potential to be applied for the detection of other pathogens.

## Materials and methods

### Isolation and maintenance of oomycete cultures

All *Saprolegnia* spp. cultures used in this study were grown and maintained on yeast peptone sucrose (YPS) agar (20 g/L D-glucose, 1 g/L KH$_2$PO$_4$, 0.5 mg/L MgSO$_4$, and 1 g/L yeast extract) supplemented with 68 μg/ml chloramphenicol and 68 μg/ml streptomycin. The cultures were incubated at 24°C in the dark [28]. Unless otherwise noted, all laboratory grade chemicals used in this study were purchased from Sigma Aldrich, St. Louis, MO.

*Saprolegnia* spp. isolates used in this study (Table 1) were isolated from various sources including fish, eggs, and waters from commercial aquaculture ponds and from recirculating

**Table 1. *Saprolegnia* spp. isolates used in the present study.**

| Organism/Sample ID | Isolation Source |
|---|---|
| *S. salmonis* | Sunshine bass eggs from Keo Fish Farm (Keo, AR). |
| 3, 9, 6A, 11A, 1B | Pond water from J.M. Malone and Sons fish farm (Lonoke, AR) |
| F1, F2, JMM Shad 3, JMM Shad 72 | American Shad (*A. sapidissima*) from ponds at J. M. Malone and Son fish farm (Lonoke, AR) |
| RAS1, RAS2, RAS3, RAS4, RAS5 | RAS water from The Freshwater Institute (Shepherdstown, WV) |

aquaculture systems (RASs). For isolation of *Saprolegnia* spp. from water, water samples from each source were collected and filtered through two layers of miracloth (Calbiochem, La Jolla, CA). One ml of the filtrate was spread aseptically onto YPS media and incubated in the dark at 24˚C. The resultant mycelial growths were subcultured for several generations to ensure the purity of the culture.

For isolation of *Saprolegnia* spp. from infected tissues, adult infected American shad (*Alosa sapidissima*) with lengths ranging between 48.3cm to 49.5cm and exhibiting visible symptoms of saprolegniasis were used for isolation of *Saprolegnia* spp following previously published protocols [29]. Briefly, areas of the fish exhibiting visible symptoms were dissected using sterilized scalpels; individual pieces were surface sterilized by treatment with 2.5% bleach for 45 s, followed by submersion in distilled water for 1 min, then in 70% ethanol for 45 s, and finally in distilled water for 1 min. These pieces were then aseptically placed onto YPS agar plates and incubated in the dark at 24˚C and observed daily.

## DNA isolation

Mycelial mats (90–100 mg) from 4-day old *Saprolegnia* cultures were collected and were crushed using a mortar and pestle in liquid nitrogen. DNA was isolated using the method of Zelaya-Molina et al. [30] with some modifications. Briefly, the crushed mycelia were treated with lysis buffer (10 mM Tris, 50 mM EDTA, 0.5% SDS, 0.5% Tween-20, and 0.5% Triton X-100), 20 mg/ml Proteinase K and 100 mg/ml RNase A. The tubes were incubated at 37˚C for 30 min in a shaking incubator at 150 rpm, vortexed for 30 s and then incubated at 55˚C for 30 min with a gentle inversion at 10-min intervals. Next, an equal volume of 25:24:1 phenol-chloroform-isoamyl alcohol (VWR, Radnor, PA) was added and the tubes were vortexed for 30 s followed by centrifugation at 10,000g for 10 min. The supernatant was transferred into a new tube, an equal volume of 24:1 chloroform-isoamyl alcohol was added, vortexed for 30 s and centrifuged at 10,000g for 10 min. The supernatant was transferred into a new tube, DNA was precipitated by adding an equal volume of isopropanol (VWR, Radnor, PA) and incubated at -20˚C for 15 min. The tubes were then centrifuged at 10,000g for 10 min and the supernatant was discarded. The resultant pellet was washed with 70% ethanol (Pharmco-AAPER, Shelbyville, KY), dried and resuspended in molecular grade water. The isolated DNA was quantified spectrophotometrically using a Nanodrop 2000c (Thermo Scientific, Wilmington, DE). Assessment of the quality of the isolated DNA was performed by electrophoresis on 0.8% agarose gels stained with SYBR™ Green I Nucleic Acid Gel stain (Invitrogen, Carlsbad, CA) and visualized using a UV transilluminator (Ultra-Lum Inc, Claremont, CA).

## Sequence analysis and phylogenetic analysis

Identification of *Saprolegnia* spp. isolates was performed by PCR of the internal transcribed spacer (ITS) region and a portion of the ribosomal large subunit [16, 31, 32] and of cytochrome C oxidase subunit 1 (CoxI) gene [33]. Amplification products were analyzed by electrophoresis and were purified using the QIAquick PCR purification kit (Qiagen, Germantown, MD) according to the manufacturer's protocol. The purified amplicons were sequenced at the University of Chicago Comprehensive Cancer Centre, DNA sequencing and Genotyping Facility (Chicago, IL). The resultant sequences were subjected to bioinformatic analysis via BLAST [34] and compared to sequences on both the NCBI [35] and the FungiDB [36] databases.

Multiple sequence alignments were generated using Clustal Omega [37]. The maximum-likelihood phylogenetic trees of CoxI and ITS markers were constructed using Mega7 software [38] based on the Tamura-Nei Model [39] with 1,000 bootstrap replicates and default

parameters. Reference ITS and CoxI sequences were obtained from NCBI GenBank database; *Phytophthora sojae* voucher P6497 and *Aphanomyces euteiches* voucher BR694 were used as the outgroups for both the trees. Concatenated phylogenetic trees of ITS and CoxI regions were constructed using SequenceMatrix version 1.8 (http://www.ggvaidya.com/taxondna/) [40] with 1,000 bootstrap replicates and default parameters.

### qPCR for specific detection of *Saprolegnia* spp.

Primers specific to the ITS region and the CoxI gene of the *Saprolegnia* genus were designed using Primer3 software with manual modifications (https://bioinfo.ut.ee/primer3-0.4.0/) [41]. Quantitative PCR reactions were carried out using the iTaq™ Universal SYBR® Green Super-mix (Bio-Rad, Hercules, CA) according to the manufacturer's instructions in a CFX Connect Real-Time PCR Detection system (Bio-Rad, Hercules, CA). To generate the standard-curve for absolute quantification, reactions were carried out using a 10-fold dilution series of *S. salmonis* DNA (ranging from 2 ng/μl to 20 ng/μl, measured spectrophotometrically). To assess the specificity of the developed qPCR, 20 ng/μl of DNA purified from all the *Saprolegnia* isolates described in Table 1 were used with both the ITS and CoxI primers. The reactions were carried out in triplicate.

### LAMP for specific detection of *Saprolegnia* spp.

To design primers for LAMP against the conserved ITS region specific for *Saprolegnia* spp., the ITS sequences for *S. parasitica* CBS 223.65 (Accession No.—PRJNA280969), *S. diclina* VS20 (Accession No.—PRJNA255245), *S. ferax* (Accession No.—PRJNA12392), *S. delica* clone 130 (Accession No.—JX212906.1) and *S. salmonis* from Argentina (Accession No.—EU551153.1) were retrieved from the NCBI sequence database [35] and aligned using Clustal Omega [37]. Inner and outer primers were generated based on this alignment, using the Primer Explorer v5 (https://primerexplorer.jp/e/, Eiken Chemical Co. Ltd., Tokyo, Japan) tool, with default parameters.

To optimize the LAMP reaction, each LAMP reaction (25 μl) contained 20 ng/μl of *S. salmonis* gDNA, 1.4 mM dNTPs, 0.32 U/ μl Bst 3.0 (New England Biolabs, Ipswich, MA), 8 mM MgSO$_4$, 1.6 μM inner primers, 0.2 μM outer primers, 0.4 μM loop primers and molecular grade water (Invitrogen, Carlsbad, CA). The reactions were incubated at 65˚C for 60 min, followed by inactivation of the enzyme to terminate the reaction at 80˚C for 5 min. No template controls (NTC) using molecular grade water served as negative controls. To each reaction, 1 μl of 1,000X SYBR Green I (AMRESCO Inc., Solon, OH) was added. Visually, a green color formation was indicative of a positive reaction whereas a golden-brown color indicated a negative reaction. The amplification products were also verified visually by 2% agarose gel electrophoresis.

### Comparison of qPCR and LAMP

A sensitivity assay was performed using a 10-fold dilution of purified *S. salmonis* DNA at 100 ng, 10 ng, 1 ng, 100 pg, 10 pg, 1 pg, 100 fg, 10 fg and 1 fg. Protocols for qPCR and LAMP assay were as described above. For detection of zoospores, one and ten zoospores were used in each reaction for both LAMP and qPCR. The reactions were performed in triplicate and repeated three times.

## Results

### Pathogen identification and phylogenetic analysis

Fifteen cultures of the oomycete were isolated from water samples as well as from infected fish tissues (Table 1). To identify the isolates, PCR was performed using both the oomycete-specific

CoxI primers and universal ITS primers. The PCR products were visualized on 0.8% agarose gel and were purified, sequenced, and analyzed by BLAST [34] with an E-value cutoff of 0. All of the isolates were identified as *Saprolegnia*. To further validate our results and to track their genetic diversity, phylogenetic trees were constructed. The maximum-likelihood tree of the ITS sequences showed that all the isolated species grouped within the *Saprolegnia* genus, in agreement with BLAST results (Fig 1). The *S. salmonis* isolate grouped closely with *S. parasitica* with a bootstrap value of 68%. For cultures isolated from the ponds of J.M. Malone and Sons fish farm (Lonoke, AR), isolates 6A and 11A were closest to *S. diclina* with a bootstrap value of 90%. Isolate 1B was grouped with *S. parasitica* with a bootstrap value of 98%. Isolate 9 grouped with *S. ferax* with 80% bootstrap support. Isolate 3 was taxonomically closest to *S. diclina* with a bootstrap value of 63%. For cultures isolated from infected fish (J.M. Malone and Sons fish farm, Lonoke, AR), isolates F1 and F2 grouped together with *S. parasitica* with 67% bootstrap support. The JMM Shad 3 and JMM Shad 72 also grouped together and appeared to have *S. parasitica* as the closest phylogenetic neighbor with 68% bootstrap support. Isolates RAS1, RAS2 and RAS3 grouped closest to *S. parasitica*, while RAS4 grouped closest to *S. ferax* with 80% bootstrap support; RAS5 had the closest phylogenetic relationship with *S. monoica* with 89% bootstrap support. A second maximum-likelihood tree based on the CoxI sequences showed the same phylogenetic groupings with the ITS based tree with slightly different bootstrap values (Fig 2). To increase discriminatory power and the robustness for identification of species within the genus, the sequences of both ITS and CoxI were used to constitute the concatenated phylogenetic tree. Fig 3 showed the same phylogenetic groupings with both the ITS- and CoxI-based trees with slightly different bootstrap values. Overall, the results of the three phylogenetic trees confirmed that the isolates in Table 1 belonged to the *Saprolegnia* genus.

## qPCR for specific detection

The qPCR reaction for CoXI gene using qCOX R1 (5'CTGAAGGACCWGAGTGHGCTTG 3') and qCOX F1 (5'GGDGCTCCWGATAGGCTTTNCC 3') was optimized by gradient qPCR. Results showed that annealing temperature at 59˚C gave the best specificity and efficiency. We then generated a standard curve using a 10-fold dilution of purified gDNA from *S. salmonis* (20 ng/µl to 0.2 pg/µl) in triplicate. The Cq values of these standards were plotted against the logarithm of their concentrations. The assay efficiency was calculated to be at 91.4% using the equation $y = (-3.546)x + 54.138$ with the slope values of -3.546 and $R^2$ values of 0.990 (Fig 4). The melt curve analysis gave rise to a single distinct peak indicating that the qPCR product was a pure, single amplicon and was specific to *S. salmonis*.

## Development of LAMP

Primers for the specific detection of *Saprolegnia* genus were designed targeting a conserved 201 bp section of the ITS region (Fig 5A). The inner and outer primers were generated based on sequence alignment using Clustal Omega alignment followed by the Primer Explorer v5 (https://primerexplorer.jp/e/, Eiken Chemical Co. Ltd., Tokyo, Japan) tool. The outer and inner primer sets with the highest dG value for dimerization and with acceptable room for generating loop primers, were chosen to ensure that the free energies and the 5' end of F1c and B1c, and the 3' end of F2 and B2, were less than or equal to -4.0 kcal/mol (Fig 5B, Table 2).

For optimization of the LAMP reaction, the first factor that was considered was the $Mg^{2+}$ concentration. We tested the reactions using 2 ng of DNA/reaction with eight different concentrations of $Mg^{2+}$ ranging from 2 mM-10 mM $Mg^{2+}$. The experiments were performed in triplicate and repeated three times. Our results showed that concentration of $Mg^{2+}$ at and

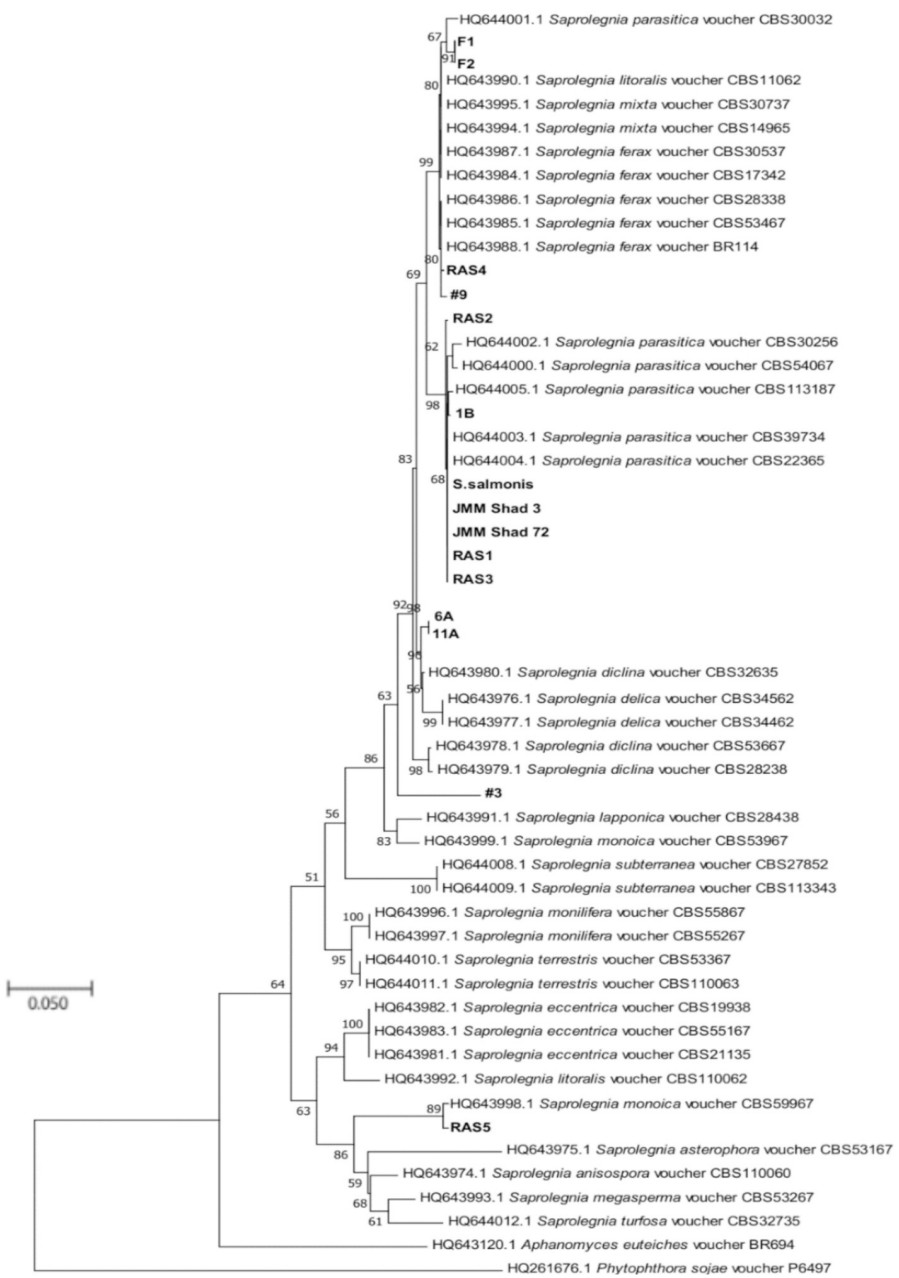

**Fig 1. Maximum-likelihood phylogenetic tree showing evolutionary relationships of 53 *Saprolegnia* isolates based on the ITS region.** Isolates examined in this study are shown in boldface. Bootstrap values ≥ 50% (1000 replicates) are given at the branchpoints. The scale bar indicates the number of substitutions per site. *P. sojae* voucher P6497 and *A. euteiches* BR694 were used as outgroups.

above 4 mM showed positive reactions. Interestingly, in no template controls, when concentrations of $Mg^{2+}$ was at 5 mM and above, the colors of the reactions showed false positives (Fig 6). Therefore, to avoid false positive reaction, the optimal concentration of $Mg^{2+}$ at 4mM was chosen. Optimization of the other factors including various temperatures, reaction times and primer ratios did not have any impact on the integrity of the reaction. Hence, the optimal

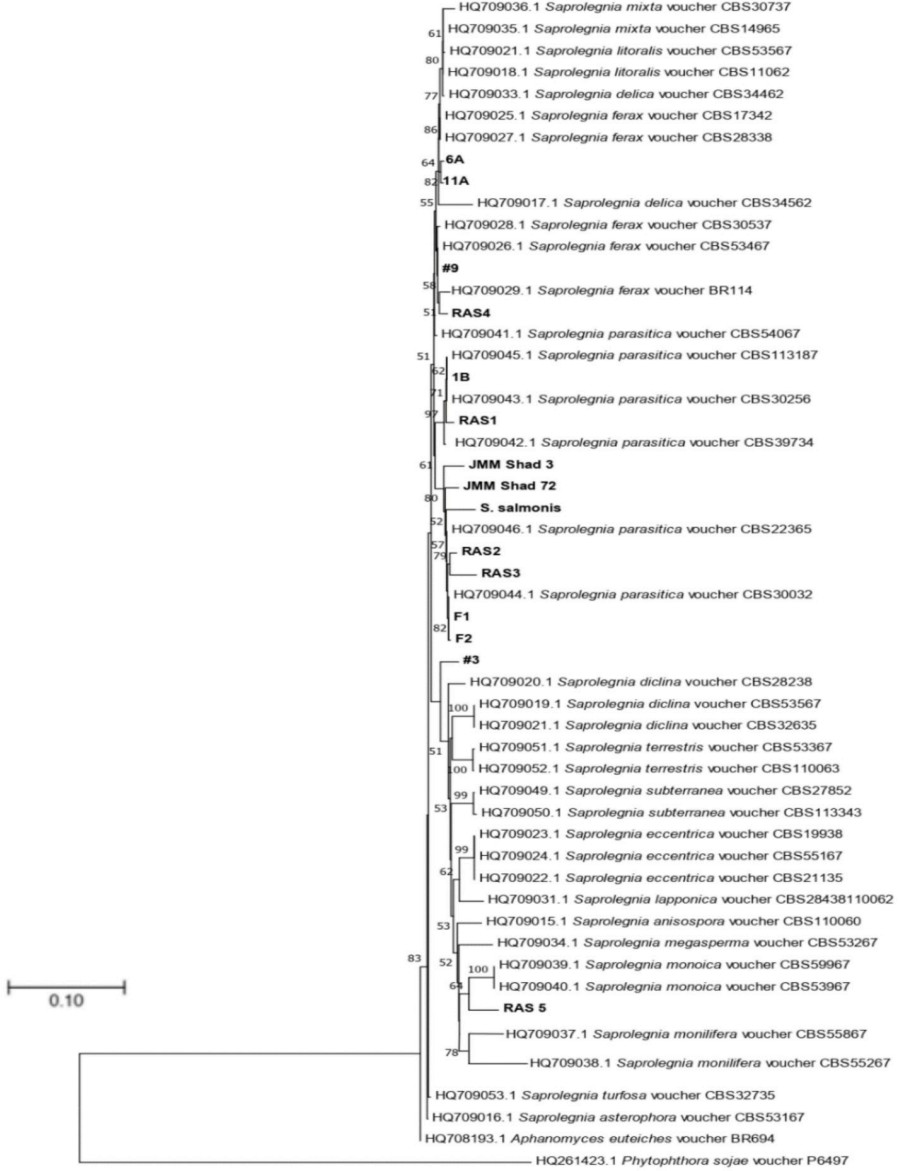

**Fig 2. Maximum-likelihood phylogenetic tree showing evolutionary relationships of 53 *Saprolegnia* isolates based on the CoxI gene.** Isolates examined in this study are shown in boldface. Bootstrap values ≥ 50% (1000 replicates) are given at the branchpoints. The scale bar indicates the number of substitutions per site. *P. sojae* voucher P6497 and *A. euteiches* BR694 were used as outgroups.

condition for the LAMP reaction was selected as 4mM $Mg^{2+}$, 1:8 outer to inner primer ratio and a reaction temperature of 65°C.

To establish the sensitivity of the LAMP reaction, the experiments were carried out with a 10-fold dilution series of purified *S. salmomis* DNA ranging from 1 fg to 100 ng, in triplicate. Results showed that LAMP successfully detected as low as 10 fg of *S. salmomis* DNA (Fig 7). This detection sensitivity was higher than that of qPCR which was at 2 pg (Fig 4). Further testing to determine whether our method can be used to directly detect zoospore from water samples without DNA purification, we subjected one or ten zoospores of *S. salmonis* to LAMP and qPCR methods. Ten ng of *S. salmonis* genomic DNA was used as the positive control and NTC

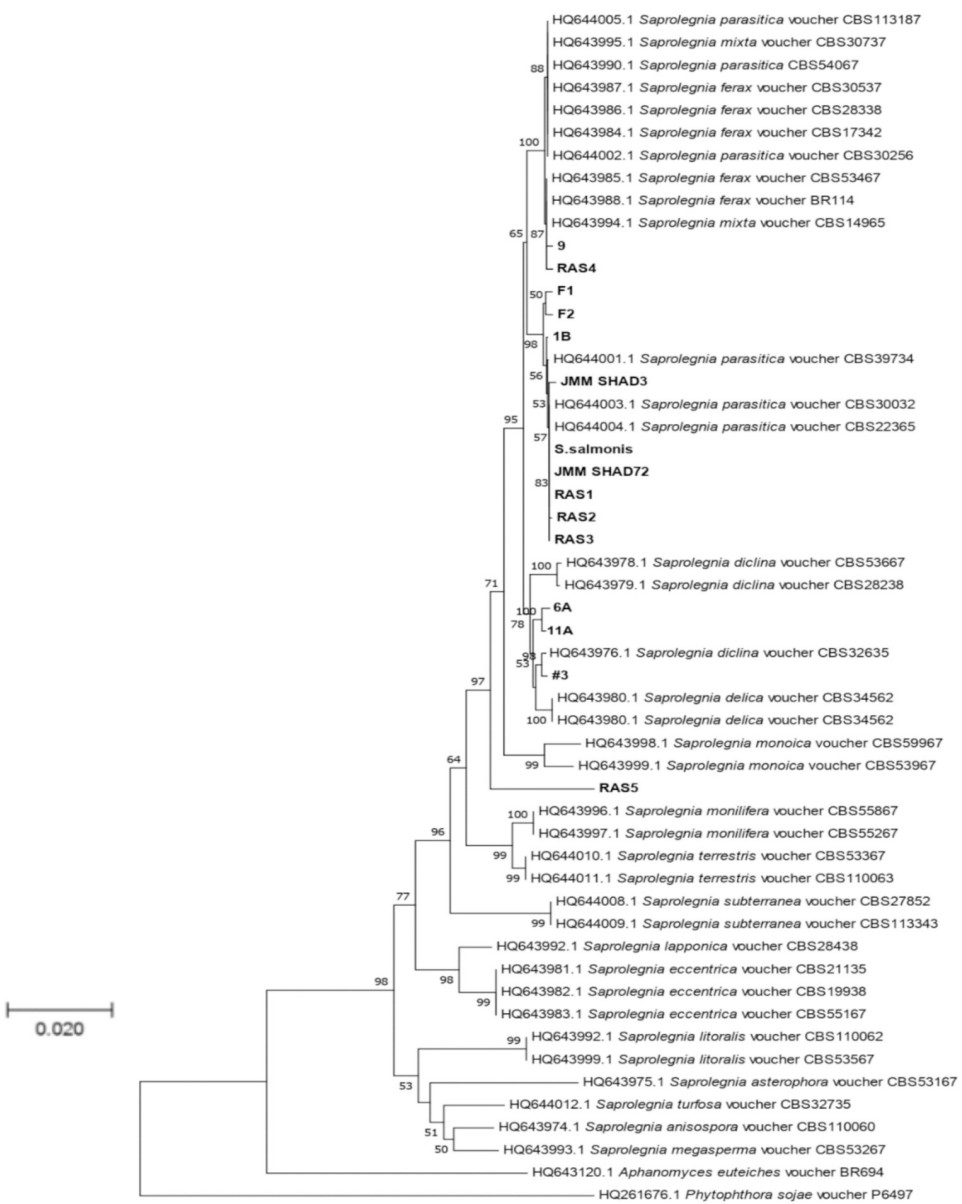

**Fig 3. Concatenation phylogenetic tree based on the sequences of ITS and CoxI.** Isolates examined in this study are shown in boldface. Bootstrap values ≥ 50% (1000 replicates) are given at the branchpoints. The scale bar indicates the number of substitutions per site. *P. sojae* voucher P6497 and *A. euteiches* BR694 were used as outgroups.

was used as negative controls. Each sample was performed in triplicate and repeated three times. The results showed that both the qPCR and LAMP procedures successfully detected one zoospore in the water samples (Fig 8), indicating that our developed methods were highly specific and highly sensitive.

The specificity of the LAMP assay was tested using purified DNA from different isolates of *Saprolegnia* spp., infected fish tissues other non-*Saprolegnia* oomycetes (*A. astaci* and *P. sojae)*, a fungus (*Fusarium* spp.), *Daphnia magna* and a bacterium (*Escherichia coli)*. Each sample was tested in triplicate. Results showed that the LAMP reaction was specific only to the *Saprolegnia*

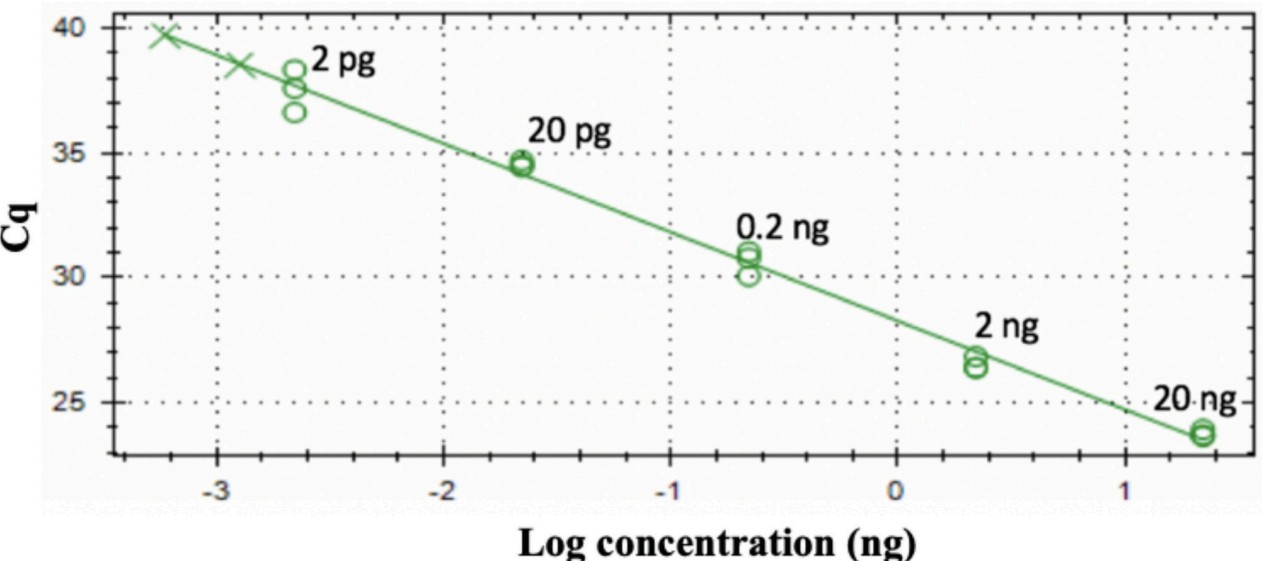

**Fig 4. Standard curve of CoxI gene.** Ten-fold dilutions of *S. salmonis* DNA ranging from 1 fg to 100 ng were amplified in triplicate and the reactions were repeated three times. The iTaq™ Universal SYBR® Green Supermix and qCOXF1+R1 primers were used.

genus (Table 3), demonstrating that our developed LAMP method was specific to pathogens in the genus *Saprolegnia* and is highly sensitive.

## Discussion

The pathogens in the genus *Saprolegnia* cause significant economic losses in aquaculture; therefore, accurate identification and early detection of pathologically relevant levels is critical. Morphological identification is challenging and less reliable compared to molecular studies. To ensure reliable identification of oomycetes, a combination of molecular markers based on ITS regions along with CoxI and/or CoxII are recommended [33]. The present study develops a standard protocol for identification of pathogens in the genus *Saprolegnia* using both ITS and CoxI. To determine genetic relationships among *Saprolegnia* spp. used in this study, phylogenetic trees with a single locus of ITS or CoxI marker as well as a concatenated phylogeny were constructed. These three phylogenetic trees confirmed that the isolates were related and belonged to the *Saprolegnia* genus. Due to low bootstrap values at some nodes, additional markers are required to conclusively determine the phylogenetic positioning of the isolates within the *Saprolegnia* genus. For instance, in all three phylogenetic trees, *S. salmonis* clustered with *S. parasitica* but bootstrap values were below 68%. Multilocus phylogenetic analysis should, in theory, increase phylogenetic resolution and improve the analysis. Such multi-gene analyses has been conducted by Göker et al. [42] using ∞-tubulin, NADH1 and LSU rDNA sequences for several oomycetes. Phylogenetic resolution of the *Phytophthora* genus was accomplished in this manner by Martin et al. [43]. Richter and Rósselló-Mora [44] suggested that measurement of average nucleotide identity between two genomes could yield more accurate results of species resolution. Finally, it has been suggested that sampling more individuals per species can help resolve incongruences amongst single gene trees [45]. An interesting observation in our phylogenetic studies was that the variation at the species level for the *Saprolegnia* isolates from RAS, even though the water samples from these systems was collected from the same tank at the same time. However, similar species level spatio-temporal variations

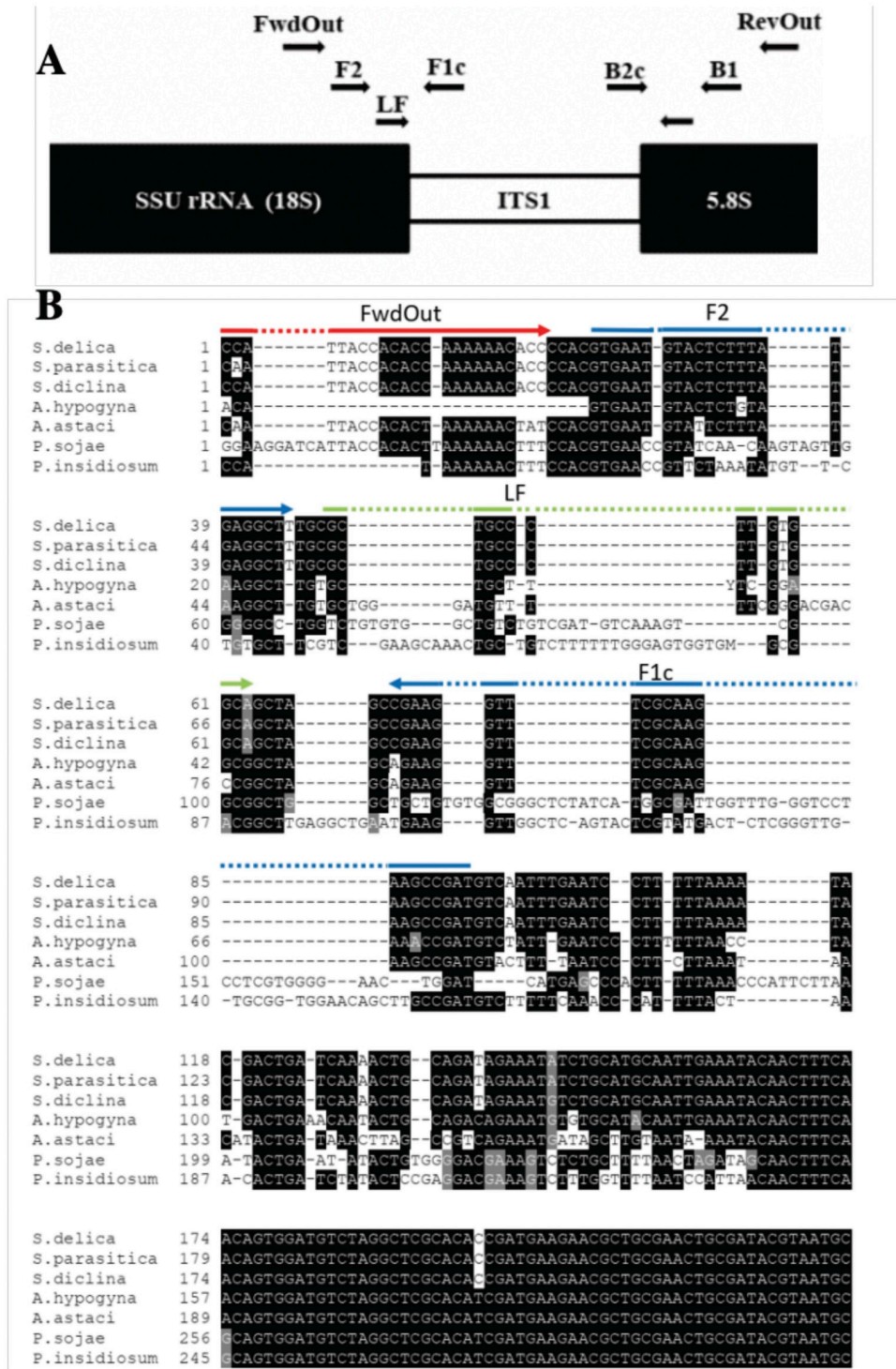

**Fig 5. Lamp primer design.** A. Schematic representation of the ITS region, showing the positioning of the LAMP primers used in the present study. Scale bar represents 100bp. FwdOut = Forward Outer Primer; RevOut = Backward Outer Primer; F2+F1c = Forward Inner Primer; B2c+B1 = Backward Inner Primer; LF = Loop Forward Primer; LB = Loop Backward Primer. B. Positioning and orientation of LAMP primers developed in the present study. Representative ITS sequences from *Saprolegnia* and non-*Saprolegnia* oomycetes were aligned and the LAMP primers were positioned on the alignment manually. The LAMP primers amplified the ITS1 regions of the target sequences.

**Table 2. LAMP primers used in the present study.**

| Primer | Abbreviation | Sequence |
|---|---|---|
| Forward Inner Primer | FIP | 5' CACTTACATGAGAAATCTCCGAA–TAGCCGAAGAACGCTTTGGAAGC 3' |
| Backward Inner Primer | BIP | 5' AATTCAGTGAGTCATCTAAATA–AACATACTCCCAGGACTAACCCGC 3' |
| Forward Outer Primer | FwdOut | 5' GGTAATGGTGTGGTTTTTTGTGG 3' |
| Backward Outer Primer | RevOut | 5' TGAAAGAAGTTTGTGTTG 3' |
| Loop Forward Primer | LF | 5' GCGACGGGAACACCGT 3' |
| Loop Backward Primer | LB | 5' AGTGCAATATGCGTT 3' |

have been reported for *Phytophthora*, *Pythium* and *Phytopythium* species in nursery irrigation systems [46].

In this study, we were able to isolate and purify *Saprolegnia* from environmental sources (water samples and infected tissues), using a similar method for the isolation of *Saprolegnia* spp. from infected rainbow trout [8], infected crayfish [47], salmon hatcheries [11] and amphibians [48]. Current protocols for detecting and identifying *Saprolegnia* from environmental sources involve lab culture-based approaches followed by PCR-based molecular identification. Another method of oomycete detection which has been used with some success is baiting [49] using rice seeds [50], hemp seeds [28] and sesame seeds [51], followed by PCR for

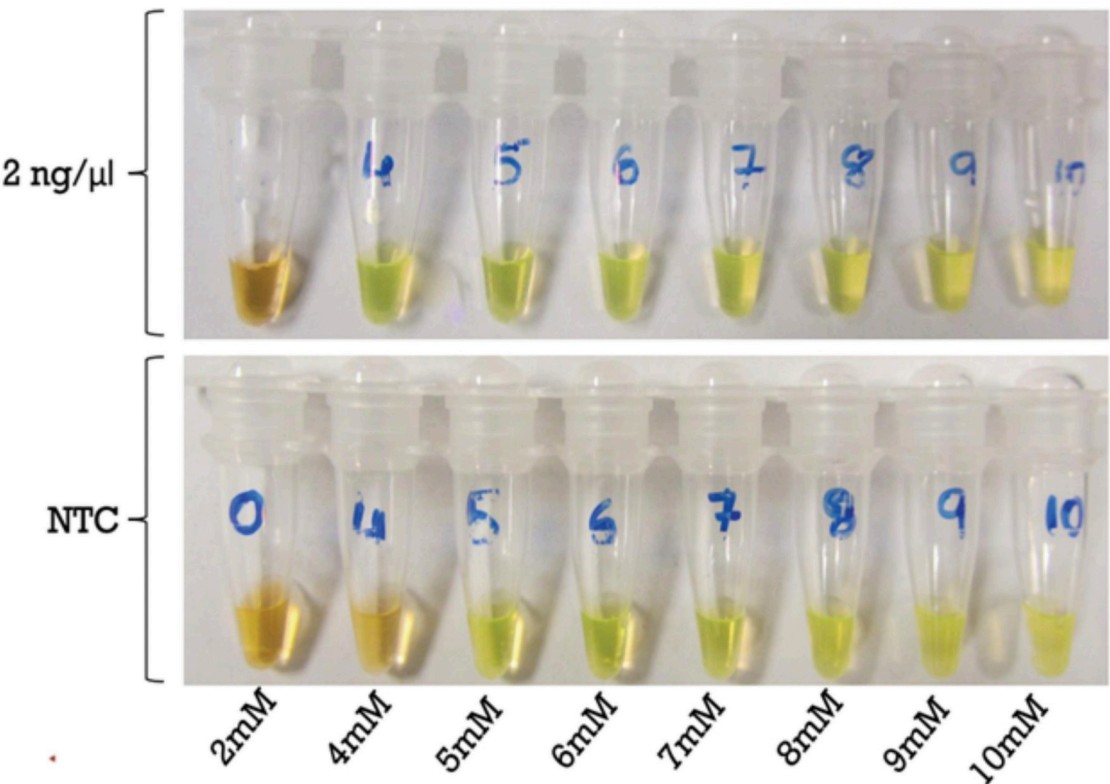

**Fig 6. Optimization of Mg²⁺ concentration for LAMP.** The concentrations of $Mg^{2+}$ ranging from 2 mm-10 mM were tested. Top panel represents positive controls using 2ng/µl *S. salmonis* gDNA. The bottom panel, No Template Control (NTC) represents negative controls. Color change to green indicates positive reaction, while golden brown indicates negative reaction. The experiments were performed in triplicate and repeated three times.

**Fig 7. The sensitivity of LAMP.** Ten-fold dilution *S. salmonis* DNA ranging from 1 fg to 100 ng were used in the presence of 1000X SYBR Green I. Color changes from golden brown to green indicates a positive reaction. The concentrations of template DNA are indicated, NTC = no template control, represents a negative control. The reactions were performed in triplicate and were repeated three times.

accurate identification. Finally, immunological detection of *Saprolegnia* spp. using monoclonal antibodies has also been reported [15].

This study is the first to develop a rapid and sensitive molecular technique for the detection of *Saprolegnia* spp. using the novel LAMP technique. Previously, LAMP has been employed for the detection of other oomycetes such as *Plasmopara viticola* [52], *P. sojae* [53] and *Pythium* spp. [54]. As an on-site detection tool, LAMP has several advantages compared to conventional PCR and qPCR. It has been applied successfully for on-site detections of the bacteria responsible for foot rot in sheep [55].

A major advantage of LAMP is that it is highly sensitive and does not require major equipment. Our developed LAMP assay successfully detected all 15 isolates in the present study and offered positive results with as low as 10 fg DNA and with only one zoospore of *S. salmonis*. It is noteworthy that our method can directly detect an organism from a sample without DNA purification, which refers to a one-step LAMP. This method has been reported in several studies such as detection of *E. coli*, *Mycobacterium smegmatis* [56] and *Mycoplasma ovipneumoniae* [57]. Similarly, detection by LAMP at femtogram level has been reported in numerous studies such as acute viral necrobiotic virus in scallop [58], classical swine flu virus [59] and *Enterococcus hirae* [60]. When our developed LAMP assay from this study was tested on several members of the oomycetes, it was found to be specific for all 15 *Saprolegnia* isolates used in this study. The sensitivity and specificity of LAMP makes it a suitable candidate for the detection of target organisms from environmental samples, especially when present in extremely low amounts. Although false positives are one of the major drawbacks associated with LAMP, much of the false positive amplification can be eliminated by using six primers and careful optimization of the reaction [61]. We did not experience any false positives in our experiments.

While both LAMP and qPCR have their own advantages and drawbacks depending on the circumstance, they can both be used to specifically detect the members of the genus *Saprolegnia*. The LAMP method is inexpensive and more rapid and sensitive compared to qPCR; therefore, we propose the use of LAMP for sensitive detection of pathologically relevant levels of *Saprolegnia* for on-site applications. However, to quantify *Saprolegnia* loads, qPCR is more useful, especially when the epidemic threshold of the disease is known. Incorporation of the LAMP method and qPCR into real-time LAMP would provide the best outcome for detection and quantification. This would enable monitoring the dynamics of *Saprolegnia* spp. in the

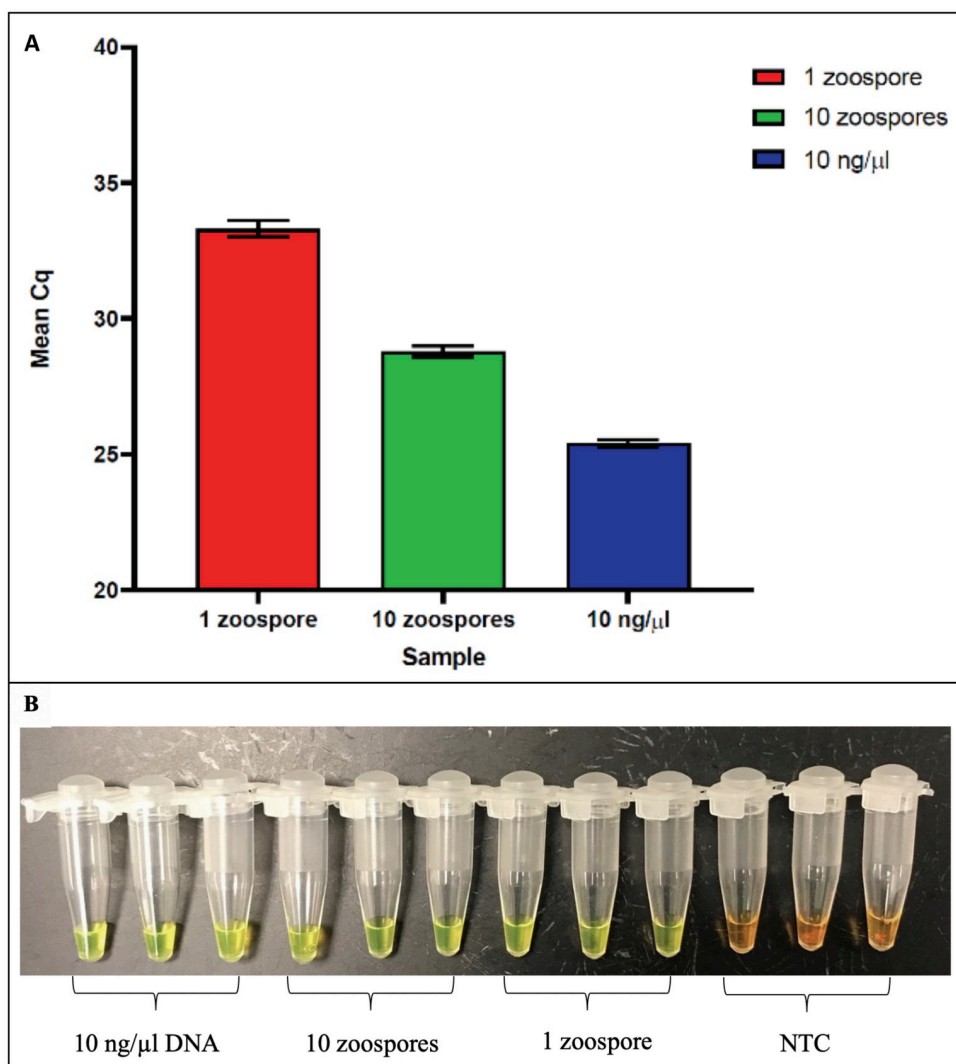

**Fig 8. Direct detection of zoospores.** A. qPCR and B. LAMP. One and ten zoospores were used in each reaction, *S. salmonis* DNA (10 ng/μl) was used as the positive control while no template control (NTC) was used as the negative control. Each sample was performed in triplicate and was repeated three times.

**Table 3. Specificity of qPCR and LAMP.**

| Species | Isolation Source | qPCR result | LAMP result |
|---|---|---|---|
| *S. salmonis* (+ control) | Pond water from J.M. Malone and Sons fish farm (Lonoke, AR) | + | + |
| *S. ferax* | Infected crayfish tissue (Bowling Green, OH) | + | + |
| *S. parasitica* | RAS water from The Freshwater Institute (Shepherdstown, WV) | + | + |
| *S. diclina* | Pond water from J.M. Malone and Sons fish farm (Lonoke, AR) | + | + |
| *S. delica* | Pond water from J.M. Malone and Sons fish farm (Lonoke, AR) | + | + |
| *A. astaci* | Infected crayfish tissue (Bowling Green, OH) | - | - |
| *P. sojae* | Infected soybean (Bowling Green, OH) | - | - |
| *Fusarium* spp. | Water from cichlid aquaria (Bowling Green, OH) | - | - |
| *D. magna* | Water from BGSU Greenhouse reservoir (Bowling Green, OH) | - | - |
| *E. coli* | TOP10 (Invitrogen, CA) | - | - |

water, and at what points their level becomes an issue. Real-time LAMP has been reported for the detection and quantification of *Enterocytozoon hepatopenaei* [62] and *Ustilago maydis* [63] and future research should investigate its development use for to detect *Saprolegnia* and other oomycete pathogens.

## Acknowledgments

We thank Gayathri Beligala for her contribution to the experiments. We thank Dr. Paul F. Morris for his technical suggestions regarding experimental design and for his comments on the manuscript. Mention of trade names or commercial products in this article is solely for the purpose of providing specific information and does not imply recommendation or endorsement by Bowling Green State University, The Conservation Fund's Freshwater Institute or the U.S. Department of Agriculture. USDA is an equal opportunity provider and employer.

## Author Contributions

**Data curation:** David L. Straus, Vipaporn Phuntumart.

**Formal analysis:** Satyaki Ghosh.

**Funding acquisition:** Vipaporn Phuntumart.

**Investigation:** Satyaki Ghosh, Vipaporn Phuntumart.

**Resources:** Christopher Good, Vipaporn Phuntumart.

**Supervision:** David L. Straus, Christopher Good, Vipaporn Phuntumart.

**Validation:** Christopher Good.

**Writing – original draft:** Satyaki Ghosh.

**Writing – review & editing:** David L. Straus, Christopher Good, Vipaporn Phuntumart.

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
