## [Decision Letter · Decision Letter 0]

6 Aug 2021

PONE-D-21-11810

Development and comparison of Loop-Mediated Isothermal Amplification with Quantitative PCR for the specific detection of Saprolegnia spp.

PLOS ONE

Dear Dr. Phuntumart,

Thank you for submitting your manuscript to PLOS ONE. After careful consideration, we feel that it has merit but does not fully meet PLOS ONE’s publication criteria as it currently stands. Therefore, we invite you to submit a revised version of the manuscript that addresses the points raised during the review process.

I got the recommendations and comments from an expert reviewer on the field. The both reviewer agree that the manuscript is technically sound and the data support the conclusions.However, lack of the real time data, such as no of technical and biological replicates..That is only one one reservation and I totally share their comments. Therefore, I can invite you to submit a revised version of the manuscript that addresses the major point raised by the reviewers.

We look forward to receiving your revised manuscript.

Kind regards,

Hideyuki Doi

Academic Editor

PLOS ONE

Journal Requirements:

“This research was funded by Ohio Sea Grant (10000960) and Bowling Green State University.  We thank Gayathri Beligala for her contribution to the experiments. “

“SG-Ohio Sea Grant (10000960), https://ohioseagrant.osu.edu/

VP-Ohio Sea Grant (10000960),

https://ohioseagrant.osu.edu/

Additional Editor Comments (if provided):

I got the recommendations and comments from an expert reviewer on the field. The both reviewer agree that the manuscript is technically sound and the data support the conclusions.However, lack of the real time data, such as no of technical and biological replicates..That is only one one reservation and I totally share their comments. Therefore, I can invite you to submit a revised version of the manuscript that addresses the major point raised by the reviewers.

Reviewers' comments:

Reviewer's Responses to Questions

**Comments to the Author**

1. Is the manuscript technically sound, and do the data support the conclusions?

Reviewer #1: Yes

2. Has the statistical analysis been performed appropriately and rigorously? 

Reviewer #1: N/A

3. Have the authors made all data underlying the findings in their manuscript fully available?

Reviewer #1: Yes

4. Is the manuscript presented in an intelligible fashion and written in standard English?

Reviewer #1: Yes

5. Review Comments to the Author

Reviewer #1: The work is indeed important for all aquaculture pathogen surveliance projects.

Good work..Indeed a good manuscipt to be publsihed.

the real time data maybe should be made more clearer, such as no of technical and biological replicates..That is only one one reservation

6. PLOS authors have the option to publish the peer review history of their article (what does this mean?). If published, this will include your full peer review and any attached files.

Reviewer #1: **Yes: **Subha Bhassu

---

## [Author Response · Author response to Decision Letter 0]

3 Nov 2021

We thank the reviewers for their helpful comments and supportive message. 

Regarding technical and biological replications in our qPCR experiment, we followed the guiding principles from Blainey, P., Krzywinski, M. & Altman, N. Replication. Nat Methods 11, 879–880 (2014). https://doi.org/10.1038/nmeth.309

We agree with the reviewers that having both technical and biological replication is of a great value. In our case, the qPCR data of Saprolegnia salmonis was relevant to assess the robustness and to quantify other Saprolegnia spp. isolates for future experiments. Additionally, Saprolegnia salmonis has been used as our major biological sample in optimization of the reactions. It was later used as a positive control in all our experiments. We hope you agree that data in Table 3 represents both technical and biological replicates for our experiments. We added this information in each of our experiments where appropriated. Thank you.

---

## [Editor Report · Decision Letter 1]

8 Nov 2021

Development and comparison of Loop-Mediated Isothermal Amplification with Quantitative PCR for the specific detection of Saprolegnia spp.

PONE-D-21-11810R1

Dear Dr. Phuntumart,

We’re pleased to inform you that your manuscript has been judged scientifically suitable for publication and will be formally accepted for publication once it meets all outstanding technical requirements.

Kind regards,

Hideyuki Doi

Academic Editor

PLOS ONE

Additional Editor Comments (optional):

I carefully checked the revised manuscript as well as the response letter. I agree the revisions according to the reviewers’ comments and now can recommend to publish the paper in this journal.
---

## [Editor Report · Acceptance letter]

3 Dec 2021

PONE-D-21-11810R1 

­­­­­­­­Development and comparison of loop-mediated isothermal amplification with quantitative PCR for the specific detection of *Saprolegnia* spp. 

Dear Dr. Phuntumart:

I'm pleased to inform you that your manuscript has been deemed suitable for publication in PLOS ONE. Congratulations! Your manuscript is now with our production department. 

Kind regards, 

on behalf of

Dr. Hideyuki Doi 

Academic Editor

PLOS ONE